# Prevalence of Health-Risk Behaviors and Mental Well-Being of ASEAN University Students in COVID-19 Pandemic

**DOI:** 10.3390/ijerph19148528

**Published:** 2022-07-12

**Authors:** Hanif Abdul Rahman, Areekul Amornsriwatanakul, Khadizah H. Abdul-Mumin, Denny Agustiningsih, Surasak Chaiyasong, Michael Chia, Supat Chupradit, Le Quang Huy, Katiya Ivanovitch, Ira Nurmala, Hazreen B. Abdul Majid, Ahmad Iqmer Nashriq Mohd Nazan, Yuvadee Rodjarkpai, Ma. Henrietta Teresa O. de la Cruz, Trias Mahmudiono, Krissachai Sriboonma, Supaporn Sudnongbua, Dhanasari Vidiawati, Apichai Wattanapisit, Sukanya Charoenwattana, Nani Cahyani, Josip Car, Moon-Ho Ringo Ho, Michael Rosenberg

**Affiliations:** 1Centre of Advanced Research (CARe), Universiti Brunei Darussalam, Bandar Seri Begawan BE1410, Brunei; hanif.rahman@ubd.edu.bn (H.A.R.); khadizah.mumin@ubd.edu.bn (K.H.A.-M.); 2School of Nursing and Statistics Online Computational Resource, University of Michigan, Ann Arbor, MI 48109, USA; 3College of Sports and Technology, Mahidol University, Nakhon Pathom 73170, Thailand; michael.rosenberg@uwa.edu.au; 4School of Human Sciences (Sport Science, Exercise and Health), University of Western Australia, Crawley, WA 6009, Australia; 5School of Nursing and Midwifery, La Trobe University, Bundoora, Melbourne, VIC 3086, Australia; 6Faculty of Medicine, Public Health and Nursing, Universitas Gadjah Mada, Yogyakarta 55281, Indonesia; denny_agustiningsih@ugm.ac.id; 7Faculty of Pharmacy, Mahasarakham University, Maha Sarakham 44150, Thailand; surasak.c@msu.ac.th; 8Physical Education & Sports Science, National Institute of Education, Nanyang Technological University, Singapore 639798, Singapore; michael.chia@nie.edu.sg; 9Department of Occupational Therapy, Faculty of Associated Medical Sciences, Chiang Mai University, Chiang Mai 50200, Thailand; supat.c@cmu.ac.th; 10School of Medicine, Vietnam National University, Ho Chi Minh City 700000, Vietnam; lqhuy@medvnu.edu.vn; 11Faculty of Public Health, Thammasat University (Rangsit Campus), Pathum Thani 12120, Thailand; katiya.i@fph.tu.ac.th; 12Health Promotion and Behavior, Faculty of Public Health, Universitas Airlangga, Surabaya 60286, Indonesia; iranurmala@fkm.unair.ac.id (I.N.); trias-m@fm.unair.ac.id (T.M.); 13Department of Social and Preventive Medicine, Faculty of Medicine, University of Malaya, Kuala Lumpur 50603, Malaysia; hazreen@ummc.edu.my; 14Faculty of Medicine and Health Sciences, Universiti Putra Malaysia, Serdang 43400, Malaysia; iqmernashriq@upm.edu.my; 15Faculty of Public Health, Burapha University, Chonburi 20131, Thailand; yuvadee@buu.ac.th (Y.R.); sukanyac@buu.ac.th (S.C.); 16Office of Health Services, Ateneo School of Medicine and Public Health, Ateneo de Manila University, Quezon City 1108, Philippines; mhdelacruz@ateneo.edu; 17Department of Teacher Training in Civil Engineering, Faculty of Technical Education, King Mongkut’s University of Technology North Bangkok, Bangkok 10800, Thailand; krissachai.s@fte.kmutnb.ac.th; 18Faculty of Public Health, Naresuan University, Phitsanulok 65000, Thailand; lillysupaporn@gmail.com; 19Faculty of Medicine, Universitas Indonesia, Depok City 16424, Indonesia; dhanasari.vt@gmail.com (D.V.); yani.sudarsono@gmail.com (N.C.); 20School of Medicine, Walailak University, Nakhon Si Thammarat 80160, Thailand; apichai.wa@wu.ac.th; 21Centre for Population Health Sciences, Lee Kong Chian School of Medicine, Nanyang Technological University, Singapore 636921, Singapore; josip.car@ntu.edu.sg; 22School of Social Sciences, Nanyang Technological University, Singapore 639798, Singapore; homh@ntu.edu.sg

**Keywords:** exercise, smoking, alcohol, diet, mental health, lifestyle habits, ASEAN

## Abstract

The prevalence of epidemiological health-risk behaviors and mental well-being in the COVID-19 pandemic, stratified by sociodemographic factors in Association of South East Asian Nations (ASEAN) university students, were examined in the research. Data were collected in March–June 2021 via an online survey from 15,366 university students from 17 universities in seven ASEAN countries. Analyzed data comprised results on physical activity, health-related behaviors, mental well-being, and sociodemographic information. A large proportion of university students consumed sugar-sweetened beverages (82.0%; 95%CI: 81.4, 82.6) and snacks/fast food daily (65.2%; 95%CI: 64.4, 66.0). About half (52.2%; 95%CI: 51.4, 53.0) consumed less than the recommended daily amounts of fruit/vegetable and had high salt intake (54%; 95%CI: 53.3, 54.8). Physical inactivity was estimated at 39.7% (95%CI: 38.9, 40.5). A minority (16.7%; 95%CI: 16.1, 17.3) had low mental well-being, smoked (8.9%; 95%CI: 8.4, 9.3), and drank alcohol (13.4%; 95%CI: 12.8, 13.9). Country and body mass index had a significant correlation with many health-risk behaviors and mental well-being. The research provided important baseline data for guidance and for the monitoring of health outcomes among ASEAN university students and concludes that healthy diet, physical activity, and mental well-being should be key priority health areas for promotion among university students.

## 1. Introduction

In the pre-coronavirus-2019 (COVID-19) pandemic era, the evidence was clear that healthy behaviors were the cornerstone to the prevention of non-communicable diseases (NCDs). NCDs have imposed a major and increasing burden on health and healthcare costs among nations [1]. Seventy-one per cent or 41 million of all global deaths (57 million) were attributable to NCDs, and a majority (78%) of all NCD deaths and 85% of global premature deaths occurred within low- and middle-income countries, where quite a few are members of the Association of South East Asian Nations (ASEAN) [2,3]. Even though many NCDs are usually asymptomatic in young adults, it is important to promote healthy, active behaviors early to prevent or delay the development of NCDs. In ASEAN, there are approximately 100 million young people (aged 15–24 years) with 20 million enrolled in a university [4]. The health behaviors of university students provide a unique forecast into future non-communicable disease (NCD) levels in later adulthood. 

The lifestyle behaviors of students at university are critical, as it is the transitionary phase from adolescence to young adulthood, and students experience many changes in life that affect all dimensions of health—intellectual, emotional, and social—and these can influence lifestyle choices that in turn affect health [5]. Many health behaviors, including tobacco and alcohol consumption, poor sleep and diet, and physical inactivity, are positioned as major modifiable risk factors to NCDs [6]. Yet, although the above-mentioned behaviors are commonly reported among university students in many regions and in many countries [7,8,9], limited research reported specifically the prevalence of these health-risk behaviors among ASEAN university students. Some literature on the lifestyle behaviors of university students are instructive. For example, large-scale studies in North America, Europe, and a multi-site study in 23 low-to-middle-income countries (LMICs) showed that between 34% and 81% of university students did not meet global physical activity (PA) guidelines [8,10,11,12,13]. It appears that a lower proportion of university students smoked (12.7–19.3%), whereas 12.2% in LMICs and 50–60% in Western countries drank alcohol heavily [8,10,12,13]. More than 80% of university students consumed insufficient fruit and vegetable daily [8,12,13]. More than one-third (35%) of university students in five ASEAN countries consumed sugar-sweetened beverages (SSBs) once or more times daily [14]. Apart from these health-risk behaviors, mental well-being is also a neglected health issue among ASEAN university students. Although previous evidence showed that mental well-being among university students in LMICs was not prevalent (12.1%) [12], the issue of mental well-being has received greater attention from educators, especially during the COVID-19 pandemic.

Attempts to promote health among university students in ASEAN has been advocated through the ASEAN University Network—Health Promotion Network (AUN-HPN). AUN–HPN was established to promote the roles of universities in health promotion, including the prevention of NCDs across the ten ASEAN countries. The network comprises more than 36-member universities across the region plus China, Japan, and South Korea [15]. The AUN-HPN Healthy University Framework developed in 2016 provides guidelines for ASEAN-member universities to adopt a holistic and comprehensive health promotion policy and programs in their institutions. Key modifiable risk factors, e.g., tobacco and alcohol consumption, poor diet, and physical inactivity, are included in the framework, as these health-risk behaviors contribute to major NCDs in the region, for instance, coronary heart diseases, diabetes, and cancers [16]. In addition to these health-risk behaviors, mental health is also emphasized as part of the advocacy [15].

However, since the establishment of the AUN-HPN and launch of the Healthy University Framework in 2018, little is known about the prevalence of major health behaviors and the factors associated with these behaviors and mental well-being in ASEAN university students. The COVID-19 pandemic and its prevention measures could exacerbate health behaviors and mental well-being of the students. Research shows that ASEAN university students experienced unprecedented levels of burdens due to the sudden partial or complete national lockdowns to contain the viral infections [17] and challenges of transitioning from physical in-person to virtual or remote learning [18,19]. Therefore, there is a cogent need for some baseline research during the COVID-19 pandemic to inform future policy, programs, and practice in the post-pandemic era of the AUN countries. To support future policies of the AUN-HPN and the implementations of the Healthy University Framework among member universities, the present study investigated the prevalence of key health-risk behaviors, mental well-being, and the sociodemographic factors associated with the behaviors and mental well-being among university students in ASEAN.

## 2. Methods

### 2.1. Study Design and Setting

The study was a cross-sectional online self-administered student survey conducted in 17 AUN-HPN member universities across 7 ASEAN countries. Participating universities included: (1) Universitas Airlangga, (2) Universitas Indonesia, and (3) Universitas Gadjah Mada from Indonesia; (4) Universiti Brunei Darussalam from Brunei Darussalam; (5) University of Malaya and (6) Universiti Putra Malaysia from Malaysia; (7) Ateneo de Manila University from the Philippines; (8) Vietnam National University Ho Chi Minh City from Vietnam; (9) Nanyang Technological University from Singapore; and (10) Burapha University, (11) Chiang Mai University, (12) King Mongkut’s University of Technology North Bangkok, (13) Naresuan University, (14) Mahasarakham University, (15) Mahidol University, (16) Thammasat University, and (17) Walailak University from Thailand. Data were collected from March to June 2021 where lockdown, movement, and social gathering restrictions were enforced in all universities due to the coronavirus outbreaks in different forms across the member countries of ASEAN.

### 2.2. Participants and Recruitment

Participating universities started the student recruitment process and distribution of the online survey at different times between March and June 2021. All undergraduate students enrolled in participating universities were invited to complete the online survey. Student recruitment and the online survey distribution were coordinated by representatives of each university and was tailored to the culture and practices of each university within each country. The recruitment methods used included public relations posters, university-wide email circulation, official university social media channels such as group LINE (Line Corporation, Tokyo, Japan), and Instagram, Facebook, and WhatsApp (WhatsApp LLC., Menlo Park, CA, USA). Students were incentivized by being eligible for a prize draw for fifty smart watches when they provided a completed survey. A link and QR code on every channel were provided for students to obtain access to the survey. 

### 2.3. Measures/Instruments

#### 2.3.1. Online Survey

The AUN-HPN online survey comprised seven sections: (1) PA, (2) social support for PA, (3) university’s environment, (4) health-related behaviors, (5) mental well-being, (6) opinion regarding university support, and (7) sociodemographic information. The survey was developed based on previously tested instruments. The survey (including all recruitment materials) originally developed in English was translated into the national language of each country. The languages included: Bahasa Indonesia (similar with Bahasa Malaysia and Bahasa Melayu), Malaysian, Thai, and Vietnamese. The translations were back translated into English according to the World Health Organization guidelines [20] to assess understandability of the questionnaire items and to rectify any inaccuracies in the translated versions. Prior to data collection, the online surveys were pilot tested with university students for comprehension of the survey and functionality of the online Qualtrics survey platform (Qualtrics International Inc., Seattle, WA, USA, 2021). Necessary changes were made, and student comments were taken into account before the rollout of the online survey. 

#### 2.3.2. Demographics

Participants provided demographic information that included age, gender, country, height, weight, grade point average (GPA), year of study, and living arrangement. Height and weight were used to calculate body mass index (BMI) and classified into 4 groups—“underweight” (<18.5 kg/m^2^), “normal” (18.5 to 22.9 kg/m^2^), “overweight” (23.0 to 24.9 kg/m^2^), and “obese” (≥25 kg/m^2^) according to World Health Organization (WHO) Asia-Pacific cut-offs [21]. GPA was standardized into a scale of 1 to 5 (≤3.2 = low, 3.3–3.9 = moderate, and >3.9 = high).

#### 2.3.3. Health-Risk Behaviors

PA was measured based on the validated Global Physical Activity Questionnaire (GPAQ) version 2.0 [22]. The 16-item GPAQ was developed and tested by WHO with acceptable concurrent validity (r = 0.54) and a high level of repeatability (0.67–0.81) [23]. PA data were expressed as weekly Metabolic Equivalent of Task unit (MET-minutes/week), which is a measure of energy expenditure, where 1 MET equals 1 kcal/kg/h [24]. Total PA levels were classified into “sufficient (≥600 MET-min/week)” and “insufficient” (<600 MET-min/week) based on the WHO PA guidelines [22]. 

Other health-risk behaviors—consumption of tobacco, alcohol, fruits/vegetables, salt, and SSBs—were estimated using items from the existing instruments [25]. For tobacco consumption, students who smoked daily were categorized into “current smokers”, and other responses (smoke occasionally/do not smoke now but used to/tried smoking a few times but never smoked regularly/have never smoked) were collapsed into “not current smokers”. For alcohol consumption, students were asked how many days in a week they usually drank alcohol. Response options ranged from 0–7 days and “do not ever drink”. Students’ drinking was classified as “daily” if they drank 7 days/week and “not daily” for responses of <7 days/week. We decided to classify students’ responses into “daily” and “not daily” drinking because generally Asians have higher alcohol-abstention rates compared with other regional groups and cultures [26]. In addition, regular alcohol drinking increased risk for all-cause and cancer mortality [27], and a zero-tolerance approach has been promoted among AUN-HPN member universities [15]. Thus, daily drinking reflected that daily consumption of alcohol could be risky. 

For fruit/vegetable consumption, students were asked how many servings of fruits/vegetables they usually ate each day (one to eight servings). The consumption was classified into “sufficient” (≥5 servings/day) and “insufficient” (<5 servings/day), according to WHO recommendations [28]. Consumption of snacks/fast food was assessed by asking how many days students ate fast food, with response options ranging from none to 7 days. Students who ate fast food every day were categorized into “daily”, and the remaining responses were collapsed into “not daily”. Students were also asked how many days they drank SSBs. Responses were handled similarly to the consumption of fast food. Frequent snacking and consumption of fast food (especially unhealthy food) are associated with higher energy intake, which could lead to higher risks of obesity [29]. Similar to snacking/fast food, frequent drinking of SSBs is associated with an increased risk of having metabolic syndromes and other NCDs, e.g., type 2 diabetes, kidney, and heart disease [30]. Thus, students’ health could be at risk with daily consumption of fast food and SSBs. Salt intake was assessed by asking how many teaspoons of salt/salty sauces were added to food before eating. Based on WHO guidelines [31], responses were categorized into “not excessive” (adding <1 tea spoon or 5 gm/per day) and “excessive” sodium intake (adding ≥1 tea spoon or 5 gm/per day).

#### 2.3.4. Mental Well-Being

Mental well-being was measured using the shortened Warwick–Edinburgh Mental Well-being Scale (WEMWBS), a reliable and valid tool for university students [32]. The scale is scored by summing responses to each item answered on a 1 to 5 Likert scale (1 = none of the time, 5 = all of the time). The minimum score is 7, and the maximum is 35. Those scored between 7.0 and 17.99 was considered as having low mental well-being, 18.0 to 24.99 was moderate, and 25.0 to 35.0 was considered as high mental well-being. 

### 2.4. Data Analysis

Data analyzed were drawn from three sections: PA, health-related behaviors, and sociodemographic information. To minimize errors in statistical analysis, a cleaning procedure was employed, such as removal of ineligible cases, duplicate responses, responses with more than 50% missing values (listwise deletion), and invalid questionnaire responses. Missing data in valid cases were handled using multiple imputation techniques set at 10 multiple imputations to replace missing with predicted values [33] using R package mice. Weighted probability was added as survey calibration to adjust and compensate for non-response bias prior to computing descriptive statistics, estimation, and inferential analyses. 

Sociodemographic characteristics, PA, mental well-being, and other health-related behaviors were described with frequency and percentage as well as binomial approximation method using 95% confidence interval. Chi-square test for independence was applied to investigate the association of sociodemographic factors and each of the study outcome variables. One-way ANOVA was used to investigate the association of sociodemographic factors with the number of health-risk behaviors (sum of all study outcomes). Logistic regressions were computed (stepwise) to examine the association between explanatory and outcome variables. All analyses were computed on R v4.1.1 and RStudio v1.4.1717 for Mac (RStudio, Boston, MA, USA). Two-sided *p*-values less than 0.05 was considered statistically significant.

## 3. Results

Table 1 presents the sample characteristics of 15,366 university students enrolled in the online survey. The majority of respondents were from Vietnam (33.3%), followed by Indonesia (28.8%) and Thailand (25.6%). Approximately, half of the respondents were female (52.6%), were 19–21 years old (66.3%), and had normal BMI (61.5%). Over half of the respondents achieved a moderate GPA of 3.3–3.9 out of 5 (69.2%) and lived off-campus (65.2%). The highest prevalence of the health-risk behaviors was daily consumption of SSBs (82.0%; 95%CI: 81.4, 82.6), followed by snacks/fast food (65.2%; 95%CI: 64.4, 66.0), low consumption of fruit and vegetable (47.8%), and having an excessive salt intake (54.0%). Insufficient PA levels (<600 MET-min/week) were observed in 39.7% (95%CI: 38.9, 40.5) of respondents. A negative level of mental well-being was observed in 16.7% of the respondents (95%CI: 16.1, 17.3), whereas 13.4% drank alcohol (13.4%; 95%CI: 12.8, 13.9) and 8.9% smoked (8.9%; 95%CI: 8.4, 9.3) (Table 2).

Table 3 and Table 4 show the proportion and significant sociodemographic factors that are supported by the final multivariable model depicted in Table 5. After adjusting for confounders, the strongest predictors of health-risk behaviors were country and BMI. University students in Indonesia and Singapore were 35% and 15%, respectively, more likely to be physically inactive compared with those in Brunei. Students in Indonesia (OR = 0.40, 95%CI: 0.34, 0.47), Malaysia (OR = 0.36, 95%CI: 0.24, 0.54), Thailand (OR = 0.36, 95%CI: 0.30, 0.42), Singapore (OR = 0.59, 95%CI: 0.42, 0.81), and Vietnam (OR = 0.48, 95%CI: 0.41, 0.56) were significantly less likely to experience low mental well-being compared with those in Brunei. Meanwhile, underweight and obese participants were 25% more likely to be physically inactive compared with those with normal BMI.

Students who lived off-campus were 28% more likely to be physically inactive and 26% more likely to have high salt intakes. Those with higher GPA were significantly less likely to consume alcohol. Female students were significantly less likely to have a poor diet (based upon fruit and vegetable intake) but were more likely to consume sugar-sweetened beverages. Higher-aged students were significantly less likely to have snack/fast food. Students in year 2 and year 4 or above were 17% and 30% more likely to have poor diet, respectively.

## 4. Discussion

The present study provided the baseline data of key health-risk behaviors and mental well-being among ASEAN university students in the COVID-19 pandemic. Several salient health-risk behaviors in university students, in particular *unhealthy diet* and *physical inactivity,* were identified behaviors, and these are instructive and helpful to the AUN-HPN stakeholders. We found that a very high proportion of ASEAN university students (82%) consumed *SSBs* daily, which were much higher than that reported in another study in university students in LMICs (35%) [12]. The frequent consumption of SSBs is worrisome, as research showed that it contributed to negative dietary patterns, including frequent fast food, high salt, and low fruit and vegetable intake [14]. Our findings also suggested that from the first year of university life onwards, students, particularly those living off campus, demonstrated increasingly poorer dietary choices. That could partially be due to a lack of self-discipline to eat a healthy diet and the fact that students had to be responsible for feeding themselves [34,35]. Additionally, the prevalence of *sufficient fruit and vegetable intake* among ASEAN university students (47.8%) was much lower compared to that reported by a multisite study in LMICs (82%) and in Australian (90%) and Canadian students (63.8%) [12,13,36]. One reason could be due to the limited access to fruits and vegetables from country-wide lockdowns and movement restrictions to prevent the spread of the COVID-19 in the seven ASEAN countries. Many traditional markets that sell affordable fruits and vegetables were closed during the pandemic. However, we recommend further research into the monitoring of the consumption of SSBs, fruit and vegetable, salt, and snack/fast food as the COVID-19 pandemic morphs into a COVID-19 endemic in seven ASEAN countries.

In the present study, *physical inactivity* among ASEAN students (39.7%) was much higher than the estimated mean for East and South-East Asia (17.3%) [37]. The prevalence of physical inactivity was, however, consistent with the 41.4% reported in a multisite study conducted in the Caribbean and South America, Sub-Saharan Africa, and East, Central, South, and South East Asia [12,37]. The results were also similar to those reported among university students in Malaysia (41.4%) [11] and Thailand (50.5%) [38]. When compared to the prevalence estimates reported in American (>70%) [39] and Canadian university students (61.2%) [36], the prevalence estimates of physical inactivity among ASEAN students were lower. Some evidence showed that the COVID-19 pandemic affected on the PA of young adults as the prevalence of meeting the PA guidelines among them decreased markedly in the COVID-19 pandemic compared to before the COVID-19 pandemic [40,41]. Nonetheless, the baseline prevalence of PA levels among ASEAN university students prior to the COVID-19 pandemic was unknown. Thus, the prevalence estimates from the present study provided useful baseline data for the monitoring of PA among university students in the event that COVID-19 pandemic morphs into an endemic disease. These results also provided the AUN-HPN with additional impetus to support its health promotion initiatives for comprehensive PA opportunities for students returning to campuses when COVID-19 restrictions are relaxed.

Several identified health-risk behaviors that were less prevalent are nonetheless still important for continuous monitoring. This included the *mental well-being* of ASEAN university students, where 16.7% of them reported low mental well-being. However, the prevalence identified in our study is markedly lower than that reported in Western countries such as Australia (33.8%) and the United States (45%) [42,43]. We propose two reasons for this lower relative prevalence of poor mental well-being in ASEAN university students. First, the prevalence could be under-reported owing to the stigma that is often associated with mental disorders in Asian cultures [44]. Second, social capital (e.g., positive family and community relationships, family support, and social interaction and support networks via social media) could have a buffering and protective function, while physical social isolation measures were operational [45].

Similarly, we suspect that the prevalence of *tobacco and alcohol consumption* might be underreported. The proportion of students who were current smokers (8.9%) in this current study was about half of the South East Asian regional estimate (19.3% among students aged ≥15 years) reported by others [46]. Moreover, the proportion of ASEAN university students who drank daily (13.4%) was much lower than that reported among people aged 15–19 years in South East Asia (21%) and in the Western Pacific regions (38%) [46]. These observed differences could partially be explained by the higher legal age requirements for alcohol and tobacco consumption in some of the ASEAN countries. (e.g., 20 years old in Thailand [47]). Additionally, some ASEAN countries, e.g., Brunei, Indonesia, and Malaysia, have populations where Muslims constitute a majority of the population and where alcohol drinking could be prohibited by the national religion. Tobacco and alcohol consumption are also prohibited in most educational institutions in the ASEAN region and might therefore contribute to the lower relative levels of consumption compared to non-ASEAN countries, where the habits of tobacco and alcohol consumption are less restrictive.

*BMI and country of residence* appeared to have a significant correlation with the lifestyle behaviors of university students. For instance, obese university students seemed to have many health-risk behaviors. They had higher odds of having insufficient PA, low mental well-being, and a higher consumption of SSBs. Similarly, students who were underweight had higher odds of not meeting the PA guidelines and having a low mental well-being. Although there was no significant relationship between BMI and snacking/fast food consumption, our results suggested that having a healthy body weight was important for university students’ well-being. Our results showed that health-risk behaviors in university students varied by the country of residence. For instance, compared to university students from Brunei, those from Indonesia and Singapore were more likely to be insufficiently active, and university students from the Philippines, Singapore, Thailand, and Vietnam had a higher chance of being daily alcohol drinkers. Similarly, university students from Malaysia, Singapore, and Vietnam had greater odds of having a high salt intake. These findings suggested that universities and public health advocates planning ameliorative health programs in different countries might need to take into account the different cultures, environments, and different priorities of the countries for effective implementation. For example, Singapore stakeholders need to pay attention to physical inactivity and alcohol and salt consumption in university students, while those in Thailand need to focus on alcohol and SSB consumption. Nevertheless, limited data in the present survey precluded a more expansive discussion. We recommend that country-level factors, e.g., urbanization, gross domestic product per capita, human development index, dietary culture, COVID-19 prevention measures, and health promotion policies and advocacy, be included in follow-up research to gain deeper insights on how these factors might play a role in the lifestyle behaviors in university students in the seven ASEAN countries.

It is noteworthy that students with a higher *GPA* had lower odds of being alcohol drinkers. The odds were even lower when the GPA was greater than 3.9. The results of our study contrasted with a study in Finland that showed that academic performance had no significant correlation with any alcohol consumption behavior (e.g., high frequency, and problematic drinking) [48]. There is, however, evidence to the contrary, where frequent drinking of alcohol is negatively associated high school and college completion [49]. As limited research on the relationship between academic performance and alcohol consumption is available, and it is unknown if the academic performance–alcohol consumption nexus relationship is bidirectional, further investigation is recommended.

Several strengths and limitations of the present research are instructive. Although this is a large-scale, multinational study that used well-defined measures and received good response from students in the midst of COVID-19 pandemic, self-reporting could underestimate certain health-risk behaviors such as tobacco, alcohol, and mental state because of social desirability bias (SDB). Nonetheless, as the online survey was anonymous, perhaps SDB was minimized. As data were collected in the COVID-19 pandemic, and responses were compounded by the COVID-19 prevention measures, statistical inference on causal and interactional events are limited with the cross-sectional nature of the research, and cause-and-effect deductions in the results cannot be ascertained. The research in the seven ASEAN countries in the COVID-19 pandemic presented opportunities for learning and fostered collaboration. Future research could use more robust prospective research designs so as to minimize biases and examine direct and indirect effects of health-risk behaviors singly or in combination (i.e., tobacco, alcohol, and other risk behaviors), environmental factors, and university policies on university students’ health behaviors.

## 5. Conclusions

Our study provided important baseline data on health-risk behaviors and mental well-being of ASEAN university students in the COVID-19 pandemic. It is conceivable that, given time, many ASEAN countries could transition to a COVID-19 endemic situation (i.e., unbridled living with COVID-19), where movement restrictions ease, and more activities would be allowed. As the students fully re-enter the academic institutions after the pandemic, healthy diet and PA should be considered as priority areas for health promotion among the AUN-HPN member universities. Additionally, conditions of poor mental well-being in university students, which are under-discussed and still stigmatized, should be carefully monitored. Existing university health promotion programs should continually be reviewed and renewed so that they can continue to stay relevant and effective for the benefit of university students across the seven ASEAN countries.

## Figures and Tables

**Table 1 ijerph-19-08528-t001:** Sociodemographic characteristics of university students (*n* = 15,366).

	Weighted		95%CI	
	*n*	%	Lower	Upper
**Age in years**, mean (SD)	20.0	1.9	19.5	20.4
18	2496	18.4	17.7	19.8
19 to 21	9016	66.3	65.9	67.5
>22	2085	15.3	14.0	16.1
**Gender**				
Female	8077	52.6	51.78	53.4
Male	7289	47.4	46.6	48.2
**BMI**				
Underweight (<18.5)	2917	21.3	19.5	23.0
Normal (18.5–22.9)	8441	61.5	58.0	65.0
Overweight (23.0–24.9)	1739	12.7	10.3	15.0
Obese (≥25)	624	4.5	2.6	7.0
**Country**				
Vietnam	5106	33.3	32.5	34.0
Indonesia	4430	28.8	28.1	3.0
Thailand	3940	25.6	25.0	26.0
Brunei Darussalam	1020	6.6	6.3	7.0
Philippines	322	2.1	1.9	2.0
Malaysia	289	1.9	1.7	2.0
Singapore	259	1.7	1.5	2.0
**Academic year**				
1st	9940	64.7	63.9	65.0
2nd	2895	18.8	18.2	19.0
3rd	1800	11.7	11.2	12.0
4th or more	731	4.8	4.4	5.1
**GPA** (1 to 5)				
≤3.2	2443	20.1	19.5	21.0
3.2 to 3.9	8406	69.2	68.1	70.5
>3.9	1302	10.7	10.2	11.3
**Living arrangement**				
Off-campus	10,021	65.2	64.5	66.0
On-campus	5345	34.8	34.0	36.0

95%CI, 95% confidence interval (maximum likelihood); SD, standard deviation; *n*, frequency; BMI, body mass index; GPA, grade point average.

**Table 2 ijerph-19-08528-t002:** Prevalence of health-related behaviors among ASEAN university students (*n* = 15,366).

	Weighted		95%CI	
	*n*	%	Lower	Upper
**Physical activity**				
Sufficient (≥600 MET-min/week)	9269	60.3	59.5	61.1
Insufficient (<600 MET-min/week)	6097	39.7	38.9	40.5
**Mental well-being**				
Low (7–17.99)	2559	16.7	16.1	17.3
Moderate (18–24.99)	9991	65.0	64.3	65.8
High (25.0–35.0)	2816	18.3	17.7	18.9
Overall score (7.0–35.0), mean (SD)	21.5	3.8	21.5	21.6
**Smoking**				
Current smokers	1365	8.9	8.4	9.3
Not current smokers	14,001	91.1	90.7	91.6
**Alcohol drinking**				
Daily	2052	13.4	12.8	13.9
Not daily	13,314	86.4	86.1	97.2
**Fruits and vegetables**				
Sufficient (≥5 servings/day)	7339	47.8	47.0	48.6
Insufficient (<5 servings/day)	8027	52.2	51.5	53.0
**Snacks/fast food**				
Daily	10,019	65.2	64.4	66.0
Not daily	5347	34.8	34.0	35.6
**Salt intake**				
Not excessive (≤5 g/day)	7061	46.0	45.2	46.7
Excessive (>5 g/day)	8305	54.0	53.3	54.8
**Sugar-sweetened beverages**				
Daily	12,598	82.0	81.4	82.6
Not daily	2768	18.0	17.4	18.6

95%CI, 95% confidence interval (maximum likelihood); SD, standard deviation; *n*, frequency.

**Table 3 ijerph-19-08528-t003:** Sociodemographic factors associated with health-risk behaviors using chi-square test (*n* = 15,366) (Frequency (percentage)).

	<600 MET	Negative MW	Smoker	Alcohol Drinker	Poor Diet	Snacking/ Fast Food	High Salt	SSB	HRB ^a^
**Gender**									
Male	**3057 (41.9)**	1190 (16.3)	**716 (9.8)**	**881 (12.1)**	**3979 (54.6)**	4724 (64.8)	3998 (54.9)	**5836 (80.1)**	3.3 (1.3)
Female	**3040 (37.6)**	1369 (17.0)	**649 (8.0)**	**1171 (14.5)**	**4048 (50.1)**	5295 (65.6)	4307 (53.3)	**6762 (83.7)**	3.3 (1.3)
**Age in years**									
18	**1124 (45.0)**	404 (16.2)	**193 (7.7)**	**162 (6.5)**	1394 (55.9)	**1644 (65.9)**	1352 (54.2)	**1989 (79.7)**	3.3 (1.3)
19 to 21	**3602 (40.0)**	1462 (16.2)	**796 (8.8)**	**1196 (13.3)**	4644 (51.5)	**5516 (61.2)**	4835 (53.6)	**7457 (82.7)**	3.3 (1.3)
>22	**754 (36.2)**	357 (17.1)	**211 (10.1)**	**361 (17.3)**	1103 (52.9)	**1300 (62.4)**	1138 (54.6)	**1755 (84.2)**	3.3 (1.4)
**Country**									
Vietnam	**1747 (34.2)**	**860 (16.9)**	**294 (5.8)**	**724 (14.2)**	**2348 (46.0)**	**1891 (37.0)**	**2902 (56.8)**	**4278 (83.8)**	**2.9 (1.3)**
Indonesia	**2437 (55.0)**	**654 (14.8)**	**440 (9.9)**	**166 (3.8)**	**2728 (61.6)**	**3473 (78.4)**	**2281 (51.5)**	**3486 (78.7)**	**3.5 (1.2)**
Thailand	**1115 (28.3)**	**562 (14.3)**	**406 (10.3)**	**996 (25.3)**	**1701 (43.2)**	**3069 (77.9)**	**2061 (52.3)**	**3378 (85.7)**	**3.4 (1.3)**
Brunei Darussalam	**437 (42.8)**	**306 (30.0)**	**136 (13.3)**	**35 (3.4)**	**682 (66.9)**	**257 (79.8)**	**549 (53.8)**	**833 (81.7)**	**3.8 (1.2)**
Philippines	**118 (36.7)**	**83 (25.5)**	**50 (15.5)**	**71 (22.1)**	**225 (69.9)**	**257 (79.8)**	**168 (52.2)**	**216 (67.1)**	**3.7 (1.5)**
Malaysia	**115 (40.0)**	**43 (14.9)**	**14 (4.8)**	**9 (3.1)**	**174 (60.2)**	**195 (79.8)**	**172 (59.5)**	**602 (69.9)**	**3.2 (1.3)**
Singapore	**128 (49.4)**	**52 (20.1)**	**25 (9.7)**	**51 (19.7)**	**169 (65.3)**	**219 (84.6)**	**172 (66.4)**	**205 (79.2)**	**3.9 (1.3)**
**Academic year**									
1st	**4081 (41.1)**	1609 (16.2)	861 (8.7)	**1154 (11.6)**	**5170 (52.0)**	**6523 (65.6)**	5402 (54.4)	8106 (81.6)	**3.3 (1.3**
2nd	**1115 (38.5)**	527 (18.2)	285 (9.8)	**497 (17.2)**	**1511 (52.2)**	**1869 (64.6)**	1559 (53.9)	2416 (83.5)	**3.4 (1.3)**
3rd	**624 (34.7)**	296 (16.4)	156 (8.7)	**304 (16.9)**	**914 (50.8)**	**1089 (60.5)**	959 (53.3)	1492 (82.9)	**3.2 (1.3)**
4th or more	**277 (37.9)**	127 (17.4)	63 (8.6)	**97 (13.3)**	**432 (59.1)**	**538 (73.6)**	385 (52.7)	584 (79.9)	**3.4 (1.4)**
**GPA**									
≤3.2	**902 (36.9)**	**405 (16.6)**	240 (9.8)	**472 (19.3)**	**1198 (49.0)**	**1478 (60.5)**	1349 (55.2)	2048 (83.8)	3.3 (1.4)
3.2 to 3.9	**3569 (42.5)**	**1302 (15.5)**	738 (8.8)	**918 (10.9)**	**4522 (53.8)**	**5555 (66.1)**	4504 (53.6)	6877 (81.8)	3.3 (1.3)
>3.9	**445 (34.2)**	**214 (16.4)**	92 (7.1)	**196 (15.1)**	**646 (49.6)**	**654 (50.2)**	727 (55.8)	1070 (82.2)	3.1 (1.4)
**BMI**									
Normal	**3313 (39.3)**	**1300 (15.4)**	**720 (8.5)**	**1097 (13.0)**	**4426 (52.4)**	**5166 (61.2)**	**4506 (53.4)**	**6888 (81.6)**	**3.2 (1.3)**
Underweight	**1286 (44.0)**	**498 (17.1)**	**169 (5.8)**	**307 (10.5)**	**1570 (53.8)**	**1749 (60.0)**	**1526 (52.3)**	**2468 (84.6)**	**3.3 (1.3)**
Overweight	**640 (36.4)**	**(308 (17.7)**	**230 (13.2)**	**249 (14.3)**	**846 (48.7)**	**1190 (68.4)**	**1025 (58.9)**	**1419 (81.6)**	**3.4 (1.3)**
Obese	**277 (44.6)**	**133 (21.3)**	**93 (14.9)**	**82 (13.1)**	**347 (55.6)**	**438 (70.2)**	**351 (56.3)**	**541 (85.7)**	**3.6 (1.3)**
**Living arrangement**									
On-campus	**1647 (30.8)**	**795 (14.9)**	503 (9,4)	**1120 (21.0)**	**2641 (46.0)**	**3667 (68.6)**	**2806 (52.5)**	**4521 (84.6)**	3.3 (1.3)
Off-campus	**4450 (44.4)**	**1764 (17.6)**	862 (8.6)	**932 (9.3)**	**5566 (55.5)**	**6352 (63.4)**	**5499 (54.9)**	**8077 (80.6)**	3.3 (1.3)

^a^ one-way ANOVA (equal variance not assumed) (Mean (Standard deviation)); Bold values = significance at < 0.05; MET, metabolic equivalent; MW, mental well-being; SSB, sugar-sweetened beverages consumption; HRB, number of health-risk behaviors (scored 0 to 8), results in Mean (Standard Deviation).

**Table 4 ijerph-19-08528-t004:** Factors associated with health-risk behaviors using bivariate logistic regression (*n* = 15,366) (Crude Odds Ratio (95%CI)).

	Physical Inactivity (<600 MET)	Negative Mental Well-Being	Smoker	Alcohol Drinker	Poor Diet	Snacking/ Fast Food	High Salt	SSB
**Age in years**								
≤18	1.00	1.00	1.00	1.00	1.00	1.00	1.00	1.00
19 to 21	**0.81 (0.74, 0.89)**	1.07 (0.92, 1.25)	1.16 (0.98, 1.36)	**2.20 (1.86, 2.62)**	**0.84 (0.77, 0.92)**	**0.82 (0.74, 0.90)**	0.97 (0.89, 1.06)	**1.21 (1.08, 1.36)**
>22	**0.69 (0.61, 0.78)**	**1.21 (1.03, 1.42)**	**1.34 (1.10, 1.65)**	**3.02 (2.74, 4.08)**	**0.89 (0.79, 1.00)**	**0.85 (0.76, 0.97)**	1.01 (0.90, 1.14)	**1.35 (1.16, 1.58)**
**Gender**								
Male	1.00	1.00	1.00	1.00	1.00	1.00	1.00	1.00
Female	**0.84 (0.78, 0.89)**	1.05 (0.96, 1.14)	**0.80 (0.72, 0.90)**	**1.23 (1.12, 1.35)**	**0.84 (0.78, 0.89)**	1.03 (0.97, 1.10)	0.94 (0.88, 1.00)	**1.28 (1.17, 1.39)**
**BMI**								
Normal	**1.00**	1.00	**1.00**	**1.00**	**1.00**	1.00	1.00	**1.00**
Underweight	**1.22 (1.12, 1.33)**	1.13 (1.01, 1.27)	**0.66 (0.55, 0.78)**	**0.78 (0.69, 0.90)**	**1.06 (0.97, 1.15)**	0.95 (0.87, 1.03)	0.95 (0.88, 1.04)	**1.23 (1.10, 1.39)**
Overweight	**0.90 (0.81, 1.00)**	1.18 (1.03, 1.35)	**1.63 (1.39, 1.91)**	**1.12 (0.96, 1.29)**	**0.86 (0.78, 0.95)**	1.37 (1.23, 1.53)	1.25 (1.12, 1.39)	**0.99 (0.87, 1.14)**
Obese	**1.24 (1.05, 1.46)**	1.49 (1.21, 1.81)	**1.88 (1.48, 2.36)**	**1.01 (0.79, 1.28)**	**1.14 (0.96, 1.34)**	1.49 (1.25, 1.79)	1.12 (0.95, 1.32)	**1.46 (1.16, 1.87)**
**Country**								
Brunei Darussalam	1.00	1.00	1.00	1.00	1.00	1.00	1.00	1.00
Indonesia	**1.63 (1.42, 1.87)**	**0.40 (0.35, 0.47)**	**0.72 (0.59, 0.88)**	1.10 (0.77, 1.61)	**0.79 (0.69, 0.92)**	**0.42 (0.33, 0.51)**	0.91 (0.79, 1.04)	**0.82 (0.69, 0.98)**
Malaysia	**0.88 (0.67, 1.15)**	**0.41 (0.28, 0.57)**	**0.33 (0.18, 0.56)**	0.90 (0.40, 1.82)	**0.75 (0.57, 0.98)**	**0.24 (0.17, 0.32)**	1.26 (0.96, 1.64)	**0.52 (0.38, 0.70)**
Philippines	0.77 (0.59, 1.00)	0.80 (0.60, 1.06)	1.19 (0.83, 1.69)	**7.96 (5.23, 12.33)**	1.15 (0.88, 1.51)	**0.45 (0.32, 0.64)**	0.93 (0.72, 1.20)	**0.45 (0.34, 0.60)**
Singapore	1.30 (1.00, 1.71)	**0.59 (0.42, 0.81)**	0.69 (0.43, 1.69)	**6.90 (4.39, 11.00)**	0.93 (0.70, 1.24)	**0.62 (0.43, 0.94)**	**1.69 (1.28, 2.26)**	0.85 (0.61, 1.20)
Thailand	**0.52 (0.46, 0.61)**	**0.39 (0.33, 0.46)**	**0.75 (0.61, 0.92)**	**9.52 (6.85, 13.69)**	**0.38 (0.33, 0.43)**	**0.40 (0.32, 0.50)**	0.94 (0.81, 1.08)	**1.34 (1.12, 1.61)**
Vietnam	**0.69 (0.61, 0.79)**	**0.47 (0.41, 0.55)**	**0.40 (0.32, 0.49)**	**4.65 (3.34, 6.69)**	**0.42 (0.37, 0.49)**	**0.07 (0.05, 0.08)**	1.12 (0.98, 1.29)	1.15 (0.97, 1.37)
**Academic year**								
1st	1.00	1.00	1.00	1.00	1.00	1.00	1.00	1.00
2nd	**0.90 (0.83, 0.98)**	**1.15 (1.03, 1.28)**	1.15 (1.00, 1.32)	**1.58 (1.41, 1.77)**	1.01 (0.93, 1.09)	0.95 (0.88, 1.04)	0.98 (0.90, 1.06)	**1.14 (1.02, 1.27)**
3rd	**0.76 (0.69, 0.85)**	1.02 (0.89, 1.17)	1.00 (0.83, 1.19)	**1.55 (1.34, 1.77)**	0.95 (0.86, 1.05)	**0.80 (0.72, 0.89)**	0.95 (0.86, 1.06)	1.09 (0.96, 1.25)
4th or more	0.88 (0.75, 1.02)	1.09 (0.89, 1.32)	0.99 (0.75, 1.29)	1.16 (0.93, 1.45)	**1.33 (1.15, 1.56**)	**1.46 (1.12, 1.73)**	0.93 (0.80, 1.08)	0.89 (0.74, 1.08)
**GPA**								
≤3.2	1.00	1.00	1.00	1.00	1.00	1.00	1.00	1.00
3.2 to 3.9	**1.26 (1.15, 1.38)**	0.92 (0.82, 1.04)	0.88 (0.76, 1.03)	**0.51 (0.45, 0.58)**	**1.21 (1.11, 1.32)**	**1.27 (1.16, 1.40)**	0.93 (0.85, 1.02)	**0.86 (0.76, 0.97)**
>3.9	0.89 (0.77, 1.02)	0.99 (0.82, 1.29)	0.70 (0.54, 0.89)	**0.74 (0.62, 0.89)**	1.02 (0.89, 1.17)	**0.65 (0.58, 0.75)**	1.02 (0.89, 1.17)	0.88 (0.74, 1.06)
**Living arrangement**								
On-campus	1.00	1.00	1.00	1.00	1.00	1.00	1.00	1.00
Off-campus	**1.79 (1.67, 1.92)**	**1.22 (1.12, 1.34)**	0.91 (0.81, 1.02)	**0.38 (0.35, 0.43)**	**1.46 (1.37, 1.56)**	**0.79 (0.74, 0.85)**	**1.10 (1.02, 1.17)**	**0.75 (0.69, 0.82)**

Bold values = significance at < 0.05; MET, metabolic equivalent; SSB, sugar-sweetened beverages consumption.

**Table 5 ijerph-19-08528-t005:** Factors associated with health-risk behaviors using stepwise multiple logistic regression (*n* = 15,366) (Adjusted Odds Ratio (95%CI)).

	Physical Inactivity (<600 MET)	Negative Mental Well-Being	Smoker	Alcohol Drinker	Poor Diet	Snacking/ Fast Food	High Salt	SSB
**Age in years**								
≤18	-	-	-	1.00	-	1.00	-	-
19 to 21	-	-	-	**1.39 (1.16, 1.68)**	-	**0.88 (0.79, 0.99)**	-	-
>22	-	-	-	**1.60 (1.29, 2.00)**	-	**0.83 (0.72, 0.97)**	-	-
**Gender**								
Male	-	-	-	-	1.00	-	-	1.00
Female	-	-	-	-	**0.92 (0.86, 0.99)**	**-**	**-**	**1.16 (1.06, 1.27)**
**BMI**								
Normal	1.00	1.00	1.00	1.00	1.00	-	1.00	1.00
Underweight	**1.25 (1.14, 1.36)**	**1.14 (1.02, 1.28)**	**0.66 (0.56, 0.78)**	**0.80 (0.70, 0.92)**	1.08 (0.99, 1.18)	-	0.96 (0.88, 1.04)	1.22 (1.09, 1.37)
Overweight	**0.86 (0.77, 0.96)**	1.13 (0.98, 1.30)	1.52 (1.29, 1.78)	1.14 (0.97, 1.33)	**0.79 (0.71, 0.88)**	-	**1.27 (1.14, 1.41)**	1.01 (0.89, 1.16)
Obese	**1.25 (1.03, 1.45)**	**1.37 (1.12, 1.68)**	1.58 (1.23, 1.99)	1.00 (0.77, 1.29)	1.01 (0.86, 1.20)	-	1.16 (0.98, 1.37)	**1.49 (1.17, 1.91)**
**Country**								
Brunei Darussalam	1.00	1.00	1.00	1.00	1.00	1.00	1.00	1.00
Indonesia	**1.65 (1.43, 1.92)**	**0.40 (0.34, 0.47)**	0.80 (0.64, 1.00)	1.00 (0.69, 1.49)	**0.82 (0.70, 0.96)**	**0.41 (0.33, 0.51)**	0.89 (0.77, 1.03)	**0.82 (0.68, 0.98)**
Malaysia	0.99 (0.73, 1.34)	**0.36 (0.24, 0.54)**	**0.36 (0.19, 0.63)**	1.14 (0.48, 2.41)	**0.68 (0.50, 0.92)**	**0.22 (0.15, 0.31)**	**1.70 (1.25, 2.31)**	**0.51 (0.37, 0.69)**
Philippines	0.75 (0.57, 0.99)	0.87 (0.64, 1.17)	1.30 (0.88, 1.89)	**7.37 (4.73, 11.65)**	1.10 (0.83, 1.47)	**0.50 (0.35, 0.72)**	1.04 (0.80, 1.36)	**0.43 (0.32, 0.59)**
Singapore	**1.85 (1.34, 2.55)**	0.73 (0.50, 1.06)	0.68 (0.38, 1.15)	**8.15 (5.03, 13.31)**	0.94 (0.68, 1.32)	**0.50 (0.34, 0.77)**	**1.98 (1.42, 2.78)**	0.88 (0.60, 1.33)
Thailand	**0.68 (0.56, 0.82)**	**0.36 (0.30, 0.42)**	0.87 (0.69, 1.09)	**8.84 (6.30, 12.81)**	**0.38 (0.32, 0.44)**	**0.40 (0.31, 0.52)**	1.19 (0.99, 1.43)	**1.40 (1.14, 1.70)**
Vietnam	**0.73 (0.63, 0.84)**	**0.48 (0.41, 0.56)**	**0.41 (0.33, 0.52)**	**4.43 (3.16, 6.42)**	**0.41 (0.35, 0.47)**	**0.05 (0.04, 0.07)**	**1.19 (1.03, 1.38)**	1.18 (0.97, 1.42)
**Academic year**								
1st	-	-	-	-	1.00	-	1.00	-
2nd	-	-	-	-	**1.17 (1.06, 1.28)**	-	0.93 (0.85, 1.02)	-
3rd	-	-	-	-	1.08 (0.96, 1.21)	-	**0.87 (0.77, 0.97)**	-
4th or more	-	-	-	-	**1.30 (1.07, 1.58)**	-	**0.81 (0.67, 0.98)**	-
**GPA**								
≤3.2	-	-	-	1.00	-	-	-	-
3.2 to 3.9	-	-	-	**0.75 (0.66, 0.86)**	-	-	-	-
>3.9	-	-	-	**0.69 (0.56, 0.84)**	-	-	-	-
**Living arrangement**								
On-campus	1.00	-	-	-	-	-	1.00	-
Off-campus	**1.28 (1.13, 1.46)**	-	-	-	-	-	**1.26 (1.11, 1.42)**	-
**H-L Goodness-of-fit test**								
χ^2^ (df)	8.66 (8)	0.862 (8)	2.14 (8)	7.08 (8)	2.34 (8)	6.14 (8)	8.51 (8)	11.00 (8)
*p*-value	0.371	0.999	0.976	0.528	0.968	0.632	0.386	0.202

Bold values = significance at <0.05; MET, metabolic equivalent; H-L, Hosmer–Lemeshow; SSB, sugar-sweetened beverages consumption.

## Data Availability

The datasets generated and/or analyzed during the current study are not publicly available due to restrictions on intellectual property regulations of the funding organization. Data are, however, available provided that an application is submitted at info@thaihealth.or.th or areekulk@gmail.com and approved by the data custodians. No administrative process is required to access the datasets.

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
