# Peer review of "Prevalence of Health-Risk Behaviors and Mental Well-Being of ASEAN University Students in COVID-19 Pandemic"

_ijerph, 2022, doi:10.3390/ijerph19148528_

Round 1
Reviewer 1 Report
Overall, the authors have provided a well-conducted and reported study across7 countries and provides a baseline for interventions and future studies. As such it makes a useful contribution to studies in higher education settings.
The methodology for the study and the survey design are transparent and can be repeated. Context is covid-19 so follow-on studies post-covid can build on this.
a few minor typos eg
line 202 an obesity should be obesity
line 205 risks should be risk
The sample size is large and on this basis the study has value for the wider research community. Limitations have been covered well.
Surprisingly mental well-being was not reported as particularly low - some context of why this could be the case would be helpful given the covid-19 restrictions. Could a discussion point on this be included?
Author Response
Please find our response to the reviewer in the attachment

Reviewer 2 Report
Authors have prepared a well written manuscript on the prevalence of health-risk behaviors and mental wellbeing of ASEAN university students, amidst the COVID-19 pandemic. The current draft reads well, and is well organized. I have several comments for authors consideration, as below.
1) In the Introduction, authors highlighted that there is a cogent need for baseline research/evidence on major health behaviors and mental well-being – would data collection period being during the peak of COVID-19 pandemic be suitable to address this gap or objective? The title of manuscript mentioned “… in COVID-19 pandemic” but in the objective (Introduction) and Conclusion there is no mention that this baseline was measured during an ongoing pandemic.
2) Background of the ongoing COVID-19 pandemic in the participating countries may provide more context on how this could impact on the investigation and outcomes of this study. Were all the countries affected equally (in magnitude of cases, burden)? What about movement restrictions enforced, and for the university students, were there cancellation of classes or adoption of virtual learning?
3) Line 90 – “mental welling”?
4) Were there reasons why the participation differs across countries? Were equal number of students approached to answer in all countries? Response rates higher in certain countries? For Singapore, Malaysia and Philippines – are the numbers (sample size) enough?
5) Were there any measures put in place to ensure that all entries were unique entries? Authors mentioned ineligible cases, duplicate responses, responses with >50% missing values and invalid responses are removed – did this affect a lot of cases?
6) GPA out of 5 was measured – do all the participating universities use the same GPA system (out of 5)?
7) Table 3 is incorrectly labelled. In Table 3, footnote mentions that bold values indicate significance <0.05. There were no bold values across the table – means all comparisons not significant?
8) Why was Brunei used as the reference group for comparison between countries?
9) Line 368: BMI and country of residence appeared to “contribute significantly” to the lifestyle behaviors of university students. It appears to suggest a causal relation that BMI contributes to the lifestyle behaviors while it could be the reverse/opposite direction, or bidirectional, which cannot be established by the observational cross-sectional design of the study. Would suggest authors to revise some of these sentences in discussion (not limited to this one), so as not to suggest causal relation.
10) Current findings, where analysis was performed for individual outcomes of interest, are sound but it may be interesting, probably in future work, that authors might want to look into how the outcomes may be related to each other. For example, physical activity, smoking and alcohol can be evaluated for their association with negative mental well-being.
Author Response
Please find our response to the reviewer's comments in the attachment
